## Article

# Proof-of-Concept Study of the Use of Accelerometry to Quantify Knee Joint Movement and Assist with the Diagnosis of Juvenile Idiopathic Arthritis

Amelia Jane Garner [1], Reza Saatchi [2,*], Oliver Ward [3], Harriet Nwaizu [2] and Daniel Philip Hawley [3]

1   The Medical School, The University Of Sheffield, Sheffield S10 2TN, UK; ajgarner2@sheffield.ac.uk
2   Industry and Innovation Research Institute, Sheffield Hallam University, Sheffield S1 1WB, UK; harriet.nwaizu@gmail.com
3   Department of Paediatric Rheumatology, Sheffield Children's Hospital, Sheffield S10 2TH, UK; oliver.ward1@nhs.ne (O.W.); daniel.hawley@nhs.net (D.P.H.)
*   Correspondence: r.saatchi@shu.ac.uk

**Abstract:** Juvenile idiopathic arthritis (JIA) is the most common rheumatic disease in childhood. Seven children and young people (CYP) with a diagnosis of JIA and suspected active arthritis of a single knee joint were recruited for this proof-of-concept study. The presence of active arthritis was confirmed by clinical examination. Four tri-axial accelerometers were integrated individually in elastic bands and placed above and below each knee. Participants performed ten periodic flexion-extensions of each knee joint while lying down, followed by walking ten meters in a straight path. The contralateral (non-inflamed) knee joint acted as a control. Accelerometry data were concordant with the results of clinical examination in six out of the seven patients recruited. There was a significant difference between the accelerometry measured range of movement (ROM, *p*-value = 0.032) of the knees with active arthritis and the healthy contralateral knees during flexion-extension. No statistically significant difference was identified between the ROM of the knee joints with active arthritis and healthy knee joints during the walking test. The study demonstrated that accelerometry may help in differentiating between healthy knee joints and those with active arthritis; however, further research is required to confirm these findings.

**Keywords:** juvenile idiopathic arthritis; accelerometry; inertia measurement unit; medical diagnosis

## 1. Introduction

This proof-of-concept study explored the use of accelerometry to objectively measure knee joint movements in children and young people (CYP) with active juvenile idiopathic arthritis (JIA). It compared accelerometry data from an actively inflamed knee with a healthy contralateral knee, thereby exploring the potential clinical application of an accelerometry device for JIA diagnosis and monitoring. In the following sections, an overview of JIA and its diagnosis, the potential use of accelerometry in JIA, and the study's methodology and results are explained.

### 1.1. Juvenile Idiopathic Arthritis

The International League of Associations for Rheumatology (ILAR) has defined JIA as arthritis of unknown etiology starting before the sixteenth birthday and lasting longer than six weeks, with all other diagnoses excluded [1]. Symptoms of active arthritis include swelling or effusion, increased warmth, and/or painful limited movement, with or without tenderness [2,3]. JIA is the most common rheumatic disease in children with an average prevalence of 70 in 100,000 in Europe [4]. The reported prevalence of JIA worldwide depends on its classification approach, geographical area, and study design, and has been

reported to be between 3.8 to 400 per 100,000 [2]. JIA more commonly affects females than males (>2:1), but this distribution varies depending on the specific disease subtype [5].

In 2001, the ILAR defined six distinct subtypes of JIA: (i) systemic arthritis, (ii) oligoarthritis, (iii) polyarthritis rheumatoid factor positive, (iv) polyarthritis rheumatoid factor negative, (v) psoriatic arthritis, and (vi) enthesitis-related arthritis [1]. However, there is increasing evidence that some of these subtypes are more heterogeneous and may be more accurately defined by a new system [6].

JIA has clinical and pathological similarities, as well as distinct differences, with adult inflammatory arthritis, known as rheumatoid arthritis (RA). RA is an inflammatory progressive disease that, if left untreated, can lead to joint destruction and disability [7]. JIA and RA have a complex genetic component involving the human leukocyte antigen (HLA) [8] but RA is a much more homogenous condition. RA tends to have worse disease outcomes compared to JIA which has variable outcomes depending on disease subtype and severity [8].

The exact cause of JIA is unknown. Possible environmental stimuli reported to be associated with JIA include infection, childhood antibiotics, maternal pregnancy smoking, gut microbes, stress, and trauma [2,9]. A genetic component is thought to be significant in JIA as monozygotic twin concordance is between 25% and 40%, showing a 250–400 times increase over population prevalence, and sibling concordance is 15–30 times above population prevalence [2].

Chronic inflammation of the joints in JIA presents as synovitis. This causes an accumulation of synovial fluid, the formation of pannus, cartilage, and bone erosions, and, consequently, joint damage [2,10]. These pathophysiological changes manifest as painful, red, swollen joints with a limited range of movement. Prolonged arthritis can lead to deformities and growth disturbances [11,12]. Joint inflammation can result in posture and movement modifications that disturb joint range of movement and change the normal walking gait [13,14]. JIA has an insidious onset which can affect its timely diagnosis and due to its close similarity to other diseases, there is a risk of misdiagnosis [9].

### 1.2. Approaches to Assist with JIA Diagnosis and Management

Approaches to aid the accurate diagnosis of JIA include a thorough clinical assessment (history and examination), appropriate blood tests, and imaging. The pediatric Gait Arms Legs Spine (pGALS) examination is a validated and easy to perform screening tool to help identify musculoskeletal abnormalities, including inflammation [10,15]. However, reliance on clinical examination alone may lead to delayed diagnoses and late commencement of treatment because changes may be subtle in the early stages of JIA and examination is inherently subjective [16].

An increase in inflammatory markers, including erythrocyte sedimentation rate (ESR) and C-reactive protein (CRP), can help support the diagnosis of JIA [17]. Blood tests are also helpful in excluding other conditions which might mimic JIA, confirming the JIA subtype and guiding treatment and management. Tests such as anti-nuclear antibodies (ANA), rheumatoid factor (RF), and anti-cyclic citrullinated peptide antibody (Anti-CCP) can aid specific JIA subtype diagnosis [2]. The information from these tests, however, is not specific to JIA and their findings should always be interpreted within the clinical context [10].

Plain radiographs are valuable in detecting certain indicators of JIA including soft tissue swelling, periarticular osteopenia, epiphyseal remodeling, and widening [10,18]. In the early stages of JIA, however, radiographs are often normal [19] and they have been shown to have low sensitivity in identifying active synovitis [17]. Ultrasound (US) can detect synovial, cartilage, and bone abnormalities earlier than conventional radiography, and may have a higher sensitivity than clinical examination [20]. However, US lacks standard references for pediatric joint imaging and it is unclear whether it can differentiate between true erosions and normal surface irregularities [21]. US scanning is also time-consuming to perform and relies on operator interpretation and experience [22]. Magnetic resonance imaging (MRI) is a highly sensitive tool for imaging joint changes associated with

inflammation [21]. Contrast-enhanced (CE) MRI can distinguish between clinically active and inactive JIA and is currently the gold-standard for diagnosing active synovitis [23]. However, MRI use is limited due to its high cost, its ability to assess only a single joint, and the requirement for sedation in younger children [21]. More recently, concerns have been raised regarding the potential accumulation of gadolinium contrast, which may further limit the use of CE-MRI as a tool to monitor inflammation in JIA [24].

Early treatment of JIA is critical to prevent permanent joint damage and its management involves a broad multidisciplinary team. This includes, but is not limited to, a consultant pediatric rheumatologist, physiotherapist, occupational therapist, ophthalmologist, psychologist, and general practitioner [17]. JIA treatment includes both pharmacological and non-pharmacological interventions managed within the context of a specialist multidisciplinary team as described above.

*1.3. Accelerometry for Joint Movement Analysis*

There is evidence that early and aggressive treatment of JIA can provide better long-term disease outcomes and this evidence indicates there could be a 'window of opportunity' in which to begin treatment [10,25,26]. Therefore, new tools, such as accelerometry, that may be able to assist in facilitating an early diagnosis could be clinically valuable. Accelerometry provides a cost-effective and quantitative means of analyzing joint movement restriction [27–29]. Accelerometry has been shown to estimate knee range of movement (ROM) to within 1% of that of goniometers [30] and has proved useful for measuring joint flexion-extension [31]. Accelerometers have also been utilized in osteoarthritis (OA) to analyze the regularity and symmetry of gait. A waist-mounted accelerometer was utilized on 15 adults with OA and 15 healthy controls and was able to accurately monitor gait regularity [32]. Accelerometry has been used to monitor physical activity in people with RA [33–36] and has been shown to identify increased vibrations in patients with RA, indicating a potential utility in RA for aiding diagnosis and monitoring [37]. Accelerometry has also been demonstrated to differentiate between normal knees and knees affected by OA, RA, and chondromalacia [38]. A later study showed RA and spondyloarthropathy could also be differentiated using accelerometry [39].

There has been limited research into the use of accelerometry in a pediatric population. This proof-of-concept study, therefore, investigates a novel application of accelerometry by examining knee joint movement in a pediatric population with JIA.

## 2. Materials and Methods

This section outlines the ethical issues, participant recruitment process, operation of the devised accelerometry device, and the data recording process and analysis.

*2.1. Ethical Approval*

The study gained research ethics approval from the Health Research Authority of Sheffield Children's Hospital, Sheffield, United Kingdom (reference 201610, date: 8 July 2020). The recruits participated in the study voluntarily and were not subjected to any harm. Participants were fully informed (including the provision of appropriate study information sheets) and appropriate study consent was sought and obtained before participation.

*2.2. Recruitment Process*

This proof-of-concept study recruited seven children who had a new or pre-existing diagnosis of JIA and current active arthritis of one knee and without suspected arthritis of any other lower limb joints (including the contralateral knee joint). COVID-19 restrictions and the specific requirements of the study inhibited the larger recruitment of ten children which was initially planned. The healthy contralateral knee acted as a reference (control) for comparison of the actively inflamed knee. Diagnosis of JIA was confirmed by an experienced pediatric rheumatologist according to the ILAR classification system [1] prior to recruitment.

A full musculoskeletal screening examination (using the pGALS tool) was performed by an experienced clinician on the day of data collection for each participant. Clinician examination confirmed the presence of active arthritis of a single knee joint and enabled the exclusion of participants suspected of having active inflammation in any other lower limb joints at the time of the study. Participants were excluded if suspected active arthritis was present in other lower limb joints as it was anticipated this may confound the accelerometry data obtained from the inflamed knee joint. The accelerometry recording was performed immediately prior to an intra-articular steroid injection of the affected knee, which is a common management strategy in JIA. Details including the JIA sub-type, age, sex, body mass index (BMI), past medical history, medical management, and the results of the screening examination were collected for each participant.

### 2.3. Participant Inclusion and Exclusion Criteria

Inclusion criteria:

- Aged $\geq 8$ and $\leq 16$ years old (this ensured participants were old enough to engage with the researchers and follow the demonstrated walking and flexion-extension movements).
- Confirmed diagnosis of JIA (polyarticular JIA, oligoarticular JIA, psoriatic JIA, and enthesitis-related JIA subtypes).
- Ability to engage with and use accelerometry equipment.

Participants were included in the study regardless of immunosuppressive therapies or other treatment options as the active treatment of JIA does not necessarily indicate disease inactivity.

Exclusion criteria:

- Aged >16 years or <8 years.
- Diagnosis of any other previous joint condition affecting the lower limbs.
- Fracture of any bone in the previous four months.
- Diagnosis of any muscle condition (e.g., muscular dystrophy).
- Diagnosis of chronic pain syndrome.
- Diagnosis of either systemic or undifferentiated JIA (this ensured homogeneity of the population sample investigated).
- Clinician or parental concern that the participant may not be able to complete accelerometry movement analysis for any reason.
- Presence of a condition that predisposes a child to any sudden, involuntary movements such as tics or chorea (these movements may interfere with accelerometer readings).
- BMI more than 99.6th centile or below 0.4th centile. The exclusion of participants with extremes of BMI controlled for the impact of skin movement artifacts on the accelerometry.

### 2.4. Materials

Four tri-axial accelerometers (type: ADXL335, acceleration measurement range: $\pm 3$ g, dimension: 4 mm $\times$ 4 mm $\times$ 1.45 mm) housed in three-dimensionally (3D) printed casings were used to obtain the accelerometry signals. The ADXL335 accelerometer can measure the static acceleration of gravity in tilt-sensing applications and dynamic acceleration from shock, vibration, or motion. Its ability to detect dynamic acceleration made it appropriate for measuring acceleration at the knee joint. The accelerometers were connected to an Arduino Mega microprocessor board, housed in a protective box. The microprocessor controlled the data recording and interfaced the sensors to a laptop computer via a USB cable. Data were collected and saved onto the laptop using the computer software CoolTerm© [40].

The accelerometer had three perpendicular axes, referred to as *x, y,* and *z,* that detected movements in three dimensions. The analog signals from the three axes of each accelerometer were sampled for conversion to discrete form and storage on the computer. The sample rate ($f_s$) was set by considering the highest frequency component of the signal being converted to discrete form. To avoid aliasing, $f_s$ needs to be at least twice the highest frequency component of the signal. The Arduino Mega microprocessor board used as

part of the recording device operated by including the data capture software in a loop, so $f_s$ was affected by the speed executed by each cycle of this loop. To measure the loop execution time, the device was allowed to operate for five minutes (*Ts*) collecting data from all four accelerometers. The number of data samples (*N*) collected per accelerometer during this interval was determined (the values for *x*, *y*, and *z* axes for an accelerometer were considered as one measure set). The device's operating $f_s$ was determined as

$$f_s = \frac{N}{Ts \times 60 \text{ s}} \tag{1}$$

Five minutes was considered long enough to give an accurate measure of $f_s$. To ensure measurements were consistent, the calculation of $f_s$ was repeated five times and the resulting values of $f_s$ were averaged. The measurements indicated the device operated at 244 samples per second. The frequency of the ROM signal was below 10 Hz and so aliasing did not occur during the discretization of the accelerometry signals. In future studies, $f_s$ could be present by controlling the analog to digital converter through software.

### 2.5. Accelerometer Data Recordings

The devised accelerometry device used in the study is shown in Figure 1.

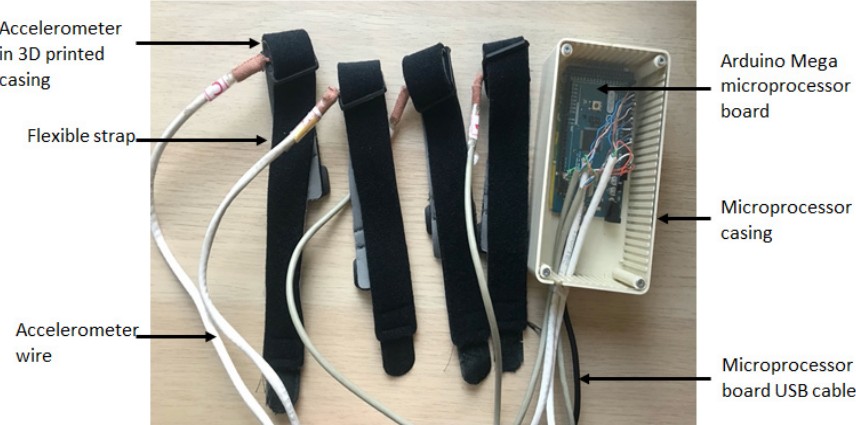

**Figure 1.** Accelerometry device used for data recordings. The microprocessor attached to the four accelerometers and flexible straps is shown.

Each accelerometer was placed inside a 3D printed casing to ensure it did not make electrical contact with the participant's body. Each casing was then integrated into a soft, easily adjustable, elastic strap to allow its attachment to the lower limb. When walking, the microcontroller was secured in a waist bag that was attached to the participant (Figure 2).

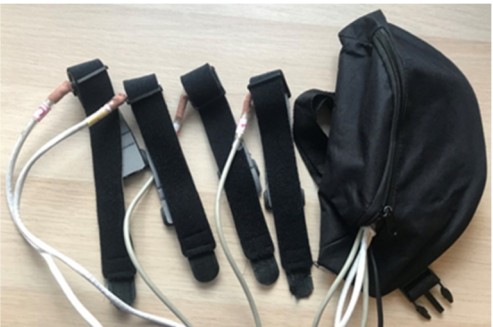

**Figure 2.** Equipment used for data collection during walking. Image of accelerometers in plastic casings, elastic straps, and microprocessor board housed in the waist bag for carrying during walking recordings.

The four accelerometers were labeled 1 to 4 and were consistently attached in the same place and in the same *x*, *y*, and *z* orientations on all participants to ensure all data

recordings were consistent. Accelerometers 1 and 2 were attached to the right leg and 3 and 4 were attached to the left leg. Accelerometers 1 and 3 were positioned above the knees, medially on the distal end of the femur. Accelerometers 2 and 4 were positioned below the knees, medially on the proximal end of the tibia. An image of the accelerometer set up for the study is presented in Figure 3.

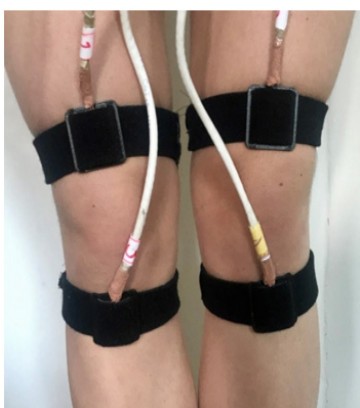

**Figure 3.** Attachment positions of the four accelerometers. Four accelerometers were attached medially to the distal femur and proximal tibia of both legs.

The selection of the medial distal end of the femur and medial proximal end of the tibia facilitated the attachment of the accelerometers to bony landmarks. This, consequently, limited the amount of accelerometer movement separate from the movement of the knee joint and so limited the amount of noise recorded.

Once the accelerometers were positioned, the participant was asked to complete the following movements whilst the accelerometry data were recorded:

- Maximally flex (flexion) and extend (extension) the leg with an actively inflamed knee ten times whilst lying down on a hospital bed. This was then repeated on the contralateral (control) side.
- Walk a distance of 10 m normally in a straight path. For this to be possible, a clinician walked with the participant, holding the PC that stored the data.

This resulted in four sets of accelerometry movement data for each participant for analysis, i.e.,

- Data from the left leg, flexion-extension movements;
- Data from the right leg, flexion-extension movements;
- Data from the left leg while walking;
- Data from the right leg while walking.

The study's protocol stated, "Individuals will be asked to walk 10 m and then asked, whilst lying, to repeatedly flex and extend each knee in turn up to 10 times or until discomfort limits movement". The repeated flexion-extension procedures were chosen in the lying down position, as this represents the body position and movement conducted during the validated pGALS musculoskeletal examination procedure. pGALS is widely published and used worldwide in the clinical setting as the standard of clinical examination. Replicating the clinical pGALS examination movement in this way ensured the study procedure was comfortable and acceptable, being a procedure that participants would already be familiar with. It also had the advantage of assessing the study intervention in the routine clinical setting in which it would potentially be used.

### 2.6. Accelerometry Data Calculations

Trigonometry and calculus operations were used to convert $x$, $y$, and $z$ signals from each accelerometer into parameters that indicated the characteristics of movement for each knee. These parameters indicated the range and period of knee movement. Detailed

descriptions of related operations and figures were described previously [41–43] and so they are very briefly outlined here. The accelerometry positioning and the angle associated with the range of movement (ROM) are as defined in a previous study [42]. ROM was determined by considering the $y$ and $z$ axes of the accelerometers, as the $x$ axis was considered parallel to the ground [42]. The knee ROM was determined using Equation (2), where $ay_1$ and $az_1$ are the readings for the $y$ and $z$ axes of the accelerometer attached to the thigh, respectively; $ay_2$ and $az_2$ are the readings for the $y$ and $z$ axes of the accelerometer attached to the shank, respectively [42]. Movements of the thigh and shank were represented by $tan^{-1}(ay_1/az_1)$ and $tan^{-1}(ay_2/az_2)$, respectively [42].

$$ROM = 180 - \left( tan^{-1}\left( \frac{ay_1}{az_1} \right) + tan^{-1}\left( \frac{ay_2}{az_2} \right) \right) \qquad (2)$$

The second variable used to analyze the knee period of movement indicated the time taken for a complete movement cycle. Matlab© package [44] was used to perform the required operations. To reduce unwanted electrical noise, the recorded accelerometry signals were filtered digitally using a fourth order Butterworth lowpass filter (cut off frequency 2 Hz).

*2.7. Data Processing and Statistical Analysis*

A typical ROM signal for a participant is shown in Figure 4. The figure includes the extent of knee movement in degrees. A small section at the start of the signal is affected by the buffering effect of digital filtering and thus was excluded from the plot and related analysis.

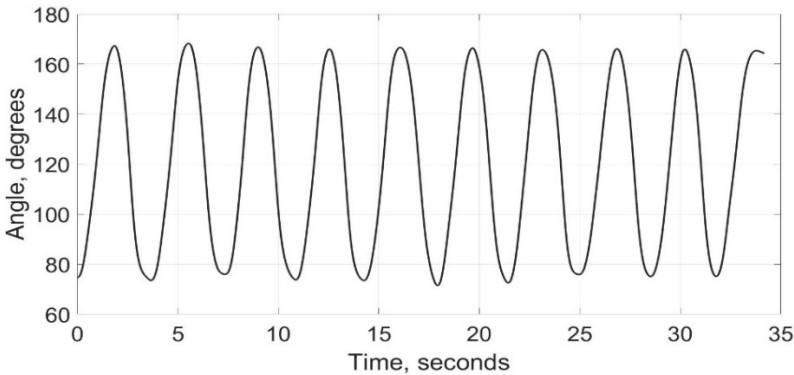

**Figure 4.** A range of movement signal.

The period of movement was determined by determining the magnitude frequency spectrum of the ROM signal. The peak in the magnitude frequency spectrum was identified ($f_p$). The period of movement was determined as $1/f_p$.

Group analysis of the whole population sample was performed using IBM SPSS© Statistics Version 26 for Mac [45]. Quantile-quantile (Q-Q) plots were used to visually examine whether the two variables were from a normal distribution. The Shapiro–Wilk test was performed to further explore whether the variables were from a normal distribution. For both variables, the Shapiro–Wilk $p$-value was greater than 0.05, thus, its null hypothesis could not be rejected, and a normal distribution was assumed. Therefore, the variables were analyzed by considering their means and standard deviations and differences were explored using paired sample $t$-tests.

Percentage difference (*PD*) and percentage absolute difference (*PAD*) were calculated for each variable between the knees with active arthritis and the contralateral healthy knees. These were calculated using Equations (3) and (4).

$$PD = \frac{healthy\ knee\ value - actively\ inflamed\ knee\ value}{healthy\ knee\ value} \times 100 \qquad (3)$$

$$PAD = absolute\left(\frac{right\ leg\ value - left\ leg\ value}{0.5 \times (right\ leg\ value + left\ leg\ value)}\right) \times 100 \qquad (4)$$

## 3. Results and Discussion

### 3.1. Participant Demographics

Tables 1 and 2 summarize the clinical demographics of the participants.

**Table 1.** Participants' JIA diagnosis and details of clinical assessment findings.

| Participant | JIA Subtype | Actively Inflamed Knee Joint | Clinician Observation | | |
|:---:|:---:|:---:|:---:|:---:|:---:|
| | | | **Warm** | **Swollen** | **Restricted** |
| 1 | Oligoarticular | Right | Yes | Yes | Yes |
| 2 | Oligoarticular | Left | Yes | Yes | Yes |
| 3 | Polyarticular | Right | No | Yes | No |
| 4 | Oligoarticular | Right | No | Yes | Yes |
| 5 | Oligoarticular | Left | No | No | No |
| 6 | Oligoarticular | Left | No | Yes | Yes |
| 7 | Polyarticular | Left | No | Yes | Yes |

**Table 2.** JIA diagnosis and details of clinical assessment for the participants.

| Characteristic | Value |
|:---|:---:|
| Sex [1] | |
|    Male | 2 (28.6%) |
|    Female | 5 (71.4%) |
| Age (years) [2] | 11.7 (2.7) |
| Body Mass Index [2] (kg/m$^2$) | 17.7 (2.7) |
| JIA subtype [1] | |
|    Oligoarticular JIA | 2 (28.6%) |
|    Polyarticular JIA | 5 (71.4%) |
| Medical management at the time of recruitment [1] | |
|    Ibuprofen | 1 (14.3%) |
|    Methotrexate and Adalimumab | 1 (14.3%) |
|    Ibuprofen and Hydroxychloroquine | 1 (14.3%) |
|    None | 4 (57.1%) |
| Knee with active arthritis at the time of recruitment [1] | |
|    Left knee | 3 (42.9%) |
|    Right knee | 3 (42.9%) |
|    None | 1 (14.3%) |

[1] Values are shown as percentages. [2] Values are shown as means (standard deviations).

### 3.2. Overall Descriptive Analysis

Patient 5 was examined on the day of the planned procedure (steroid injection) and found to have no signs of active arthritis in either knee joint. Accelerometry data showed no significant difference in either joint (and was therefore concordant with clinical examination); because patient 5 showed no signs of active knee arthritis at the time of the accelerometry recording, this patient was excluded from all further analyses. The following group analysis, therefore, presents data from the remaining six participants. Table 3 provides summary statistics for the accelerometry data collected. The Percentage Difference (PD) and Percentage Absolute Difference (PAD) between the values of the healthy knee and actively inflamed knee are also presented.

**Table 3.** Summary statistics for knee range of movement and period of movement, and the concordance with pGALS examination findings.

| Accelerometry Variable | Flexion-Extension | | | Walking | | |
| --- | --- | --- | --- | --- | --- | --- |
| | Actively Inflamed Knee | Healthy Knee | PD (PAD) | Actively Inflamed Knee | Healthy Knee | PD (PAD) |
| Range of knee movement (degrees) [1] | 88.34 (16.63) | 105.59 (10.49) | 16.3% (17.8%) | 28.50 (3.62) | 32.76 (4.92) | 13.0% (13.9%) |
| Period of movement (s) [1] | 3.38 (0.75) | 3.32 (0.49) | −1.8% (1.8%) | 1.32 (0.21) | 1.33 (0.21) | 0.8% (0.8%) |
| Concordance with pGALS examination finding2 | | | | | | |
| Yes | | 5 (83%) | | | 4 (67%) | |
| No | | 1 (17%) | | | 2 (33%) | |

[1] Results presented are mean (standard deviation).

The results presented in Table 3 and Figure 5 indicate that the ROM was reduced in the knees with active inflammation compared to the healthy contralateral knee in both movement modalities (i.e., flexion-extension and walking). For ROM, the percentage difference (PD) was lower by 16.3% and 13.0% in the knees with active arthritis during flexion-extension and walking movements, respectively. Using percentage absolute difference (PAD), ROM showed differences of 17.8% and 13.9% during flexion-extension and walking movements, respectively, when examining the knees with active arthritis and the corresponding contralateral. For the period of movement, PD values were higher by 1.8% and 0.8% in the knees with active arthritis during flexion-extension and walking movements, respectively. Using PAD, this variable gave comparable results.

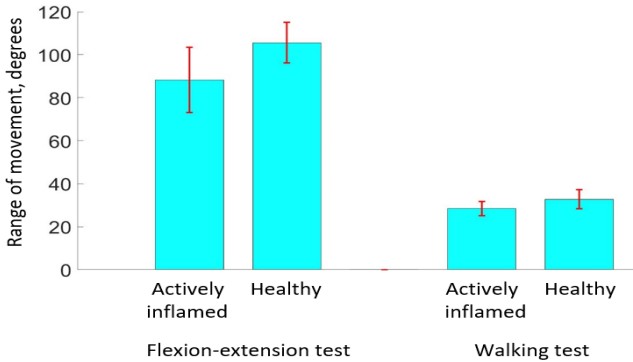

**Figure 5.** Mean ROM for flexion-extension and walking tests. The red lines on the top of the bars are indicators of associated standard deviations.

*3.3. Tests for Significance*

As both ROM and period of movement variables were from a normal distribution, paired sample *t*-tests (confidence interval 95%) were performed to explore significant differences. These results indicated that for ROM during flexion-extension, there was a significant difference between the actively inflamed and healthy knees ($p = 0.032$). This difference was not statistically significant for the walking test. For the period of movement, the differences were not statistically significant for both flexion-extension and walking tests.

**4. Discussion**

The analysis confirmed that there was a statistically significant difference between the ROM accelerometry data collected from knees with active inflammation and healthy knees during flexion-extension. There was concordance between the clinical diagnosis and accelerometry findings in six out of the seven participants included in the study.

The developed accelerometry device is safe, cost-effective, easy to use, portable, and provides accurate quantification of knee joint movement. As joint restriction is a feature of JIA, this measurement can have potential in both diagnosis and monitoring of patients

affected by it. Diagnosis of active arthritis in JIA currently relies on clinical judgment based on multiple sources of information including history, examination, and findings from investigations. Accelerometry data sits within this context. Table 4 presents a summary of the relationships between the accelerometry results.

**Table 4.** Summary of the relationship between accelerometry recordings.

| Variable | Accelerometry Results | | Significance |
|---|---|---|---|
| | **Flexion-Extension** | **Walking** | |
| Range of movement (degrees) | Reduced in actively inflamed knees in 83.3% of cases (mean percentage difference 16.3%) | Reduced in actively inflamed knees in 83.3% of cases (mean percentage difference 13.0%) | Significant during flexion-extension (*p*-value = 0.032) Not significant during walking |
| Period of movement (seconds) | Greater in actively inflamed knees in 66.7% of cases (mean percentage difference 1.8%) | Similar in both knees in all cases (mean percentage difference 0.8%) | Not significant |

*Strengths and Limitations of the Study*

Our proof-of-concept study had a few strengths and limitations which we will address here. The use of accelerometry in the medical field has been researched previously, however, there have been few studies on its applications in JIA. One of the key strengths of this study is that it explored a potentially valuable application of accelerometry which, to the best of our knowledge, has been relatively unexplored. There has been extensive research on the use of accelerometry to quantify the physical activity of patients with arthritis in both adult and pediatric populations [33,36,46,47]. There have also been studies investigating joint movement more specifically in adult populations with RA [38,39]. These studies found that accelerometry was able to objectively assess joint movement and could aid the differentiation of different types of joint disease [39]. This study confirmed accelerometry may also be valuable in assisting with the assessment of a pediatric population with arthritis. This study importantly provides evidence that the promising use of accelerometry in adult populations may also be transferable to pediatric populations with JIA. This proof-of-concept study has provided evidence to support further research into accelerometry use in JIA.

A further specific strength of this study was the collection of accelerometry data during both walking and flexion-extension movements. In assessing proof-of-concept, this allowed us to explore two potential applications of this technology simultaneously. Our study supported results from other studies exploring gait kinematics [14], however, no studies were identified that also explored individual knee joint analysis from single leg joint flexion-extension. We specifically chose to explore this method of data recording for several reasons. It allowed data to be collected independently from the contralateral leg, therefore, the movement of the joint being recorded was not influenced by the movement of the contralateral joint. In our study, this method allowed greater differentiation between knees with active arthritis and healthy knees during flexion-extension as opposed to a single recording of walking data being collected.

A third specific strength of our study was the exclusion of participants with suspected active arthritis in any lower limb joints other than the knee being assessed. Active arthritis in a hip joint or an ankle joint may affect how the knee joint moves and could therefore interfere with the accelerometry data of knee movements. In this way, we reduced the potential for arthritis in other joints (e.g., hips or ankles) to confound the accelerometer data we were collecting by comparing a healthy and actively inflamed knee joint.

Our study also had a few limitations. Firstly, the identification of the joint with active arthritis relied on clinician assessment alone. Contrast-enhanced (CE) MRI scanning is considered the 'gold-standard' assessment for active synovitis, and future work could compare accelerometry with CE-MRI. In this proof-of-concept study, however, we sought to explore the utility of accelerometry within a usual clinical setting. Clinician examination (using pGALS tool) is the usual method for assessing active arthritis in JIA because of

the limitations of CE-MRI (cost, accessibility, and the need for gadolinium contrast) as described above.

A second limitation of our study was that individuals with multiple inflamed lower-limb joints were excluded. Whilst this method allowed us to obtain clear accelerometry data in relation to a single inflamed knee, we were not in a position to comment on how multiple inflamed lower-limb joints might be reflected in accelerometry data. This could be looked at as part of a larger study. Future work could also explore accelerometry data in other inflamed joints in individuals with JIA.

A third limitation of our study was the reliance on a clinician-assessed healthy knee joint to provide a normative comparison with the clinician-assessed inflamed joint. Clinician assessment may miss subclinical active arthritis and it is, therefore, possible that some of the clinician-assessed healthy knees in the study may have been affected by sub-clinical active arthritis. Future work could obtain normative accelerometry data from a population of children with no known joint disease.

Our study aimed to recruit a larger number of participants to describe the proof-of-concept. The challenges posed by the COVID-19 pandemic and the study's strict exclusion criteria resulted in a final sample size of seven (of which six data sets were fully analyzed).

## 5. Conclusions

A proof-of-concept study was carried out to investigate the application of accelerometry to quantitatively examine knee joint movement in individuals with JIA. The study explored accelerometry variables that suitably characterized the joint movement. The range of movement obtained by accelerometry in the flexion-extension test was found to be lower in actively inflamed knees as compared with healthy contralateral knees. The difference was statistically significant. It was found that accelerometry has the potential to differentiate between actively inflamed and healthy knees. The method has also the potential for monitoring JIA. The novel aspects of the study include the examination of joint movement in JIA and, specifically, the combined exploration of both flexion-extension and walking examinations. Further research is required, on larger sample size, to confirm these findings and refine the use of this novel technology in children and young people with JIA.

**Author Contributions:** Conceptualization, A.J.G., R.S., O.W., H.N. and D.P.H.; methodology, A.J.G., R.S., O.W., H.N. and D.P.H.; validation, A.J.G., R.S., O.W., H.N. and D.P.H.; investigation, A.J.G., R.S., O.W., H.N. and D.P.H.; data curation, A.J.G., R.S., O.W., H.N. and D.P.H.; writing—original draft preparation, A.J.G., R.S., O.W., H.N. and D.P.H.; writing—review and editing, A.J.G., R.S., O.W., H.N. and D.P.H. All authors have read and agreed to the published version of the manuscript.

**Funding:** This research received no external funding.

**Institutional Review Board Statement:** The study was conducted in accordance with the Declaration of Helsinki, and approved by Health Research Authority, Sheffield Children's Hospital, Sheffield, United Kingdom (Reference: 201610, date 8 July 2020).

**Informed Consent Statement:** Informed consent was obtained from all subjects involved in the study.

**Data Availability Statement:** Due to ethical restrictions the study's data will not be shared.

**Acknowledgments:** The authors are very grateful to all the children who took part in the study and for the cooperation of their caregivers.

**Conflicts of Interest:** The authors declare no conflict of interest.

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
