# Peer review of "Proof-of-Concept Study of the Use of Accelerometry to Quantify Knee Joint Movement and Assist with the Diagnosis of Juvenile Idiopathic Arthritis"

_technologies, doi:10.3390/technologies10040076_

Round 1

Reviewer 1 Report

In this paper, a proof-of-concept study of the use of accelerometry to quantify knee joint movement and assist with the diagnosis of juvenile idiopathic arthritis was carried out.  The topic of the paper was interesting and the paper was well organized. Followings are comments for this paper.

  1. Figure 7 shows the result of total angular velocity, and the unit is "m" in the graph. Also on page 7, line 28, the angular displacement is defined with the unit of "m. However, in general, angular displacement is expressed in units of degree. In order to help understand what the variables used in this paper mean, it would be good to explain the ROM, acceleration, velocity, and angular displacement defined on page 7 using graphics.
  2. In the results of Figures 5, 6, and 8, only the result of actively inflamed knee during flexion-extension of patient 4 shows a different trend. An explanation is needed as to the cause of such an result.
  3. There is no mention of Figure 7 in the manuscript, and there is no analysis of the results. The content of Figure 7 should be added to the text.

Author Response

Dear Honorary editor, reviewer

Thank you very much for so kindly reviewing our paper and making very valuable constructive comments. We have considered all your comments carefully and have incorporated them into the revised paper, significantly enhancing it. The changes made are summarised in the attached document and highlighted on the revised paper. We have also proofread the paper again.   

We are very grateful for the help and support provided and hope our revisions meet your expectations.

Best wishes

Professor Reza Saatchi

Reviewer 2 Report

The paper presented a study to use the accelerometer to measure the motion of the joint as the potential diagnostics of the juvenile idiopathic arthritis. The idea is interesting but the method in the study was not rigorous.

The study used the flexion-extension angle, velocity and acceleration of the leg. The velocity and acceleration measurement were questionable. Were the patient asked to swing the leg or walk as fast as they could? The same action can be done within different time frame. Thus those values were arbitrary and unreliable. The velocity and acceleration are angular velocity and acceleration. Their units should be s-1 and s-2 based on equations (5) and (6). They are also derivatives of the angles, which usually amplifies the noise in the signal. How did the noise affect the accuracy of the maximum speed and acceleration? Those values should not be used unless the experiment protocol can ensure they are the "maximums".

Equation 7 is the reverse of equation 5 so the displacement should be the αflexion_extension. Why was it recalculated?

The sample number is small. How does the sample size affect the outcome of the study? The data only showed that there could be different outcomes between the leg with JIA and the leg without JIA from the same subject. But it did not answer the question if there would be a threshold to separate JIA patient from a non JIA patient.

The sample rate was 244 s/s. Why was this number used? Why not 250s/s or 200s/s?

Please provide a figure to show the angles ??â„Ž??â„Ž and ??â„Ž??? with respect to the arrangement of the accelerometers?

Please provide an example of the measured signals.

Please correct the color scheme in the figure 4. The colors should be correct and consistent. The standard deviations should be the error bars on the mean values. What was the purpose of the medium and IQR values? They should be removed.

Author Response

Dear Honorary editor, reviewer

Thank you very much for so kindly reviewing our paper and making very valuable constructive comments. We have considered all your comments carefully and have incorporated them into the revised paper, significantly enhancing it. The changes made are summarised in the attached document and are highlighted on the revised paper. We have also proofread the paper again.   

We are very grateful for the help and support provided and hope our revisions meet your expectations.

Best wishes

Professor Reza Saatchi

Round 2

Reviewer 2 Report

The revision did not address the concerns in the previous review.

The use of ROM as the criteria is reasonable. However, the patients can extend the legs at the velocity that patient feels comfortable based on the authors' response. There was no requirement to move the leg as fast as possible. So the velocity is arbitrary. The maximum velocity is not the true maximum velocity and is not comparable. This is true for acceleration. Thus the velocity and acceleration data should be removed. 

It is a statistical flaw to assume that 10 is the reasonable number of subjects. The correct procedure is to conduct a statistical power analysis to determine the number of the subjects. 

The determination of the sample frequency is not correct. It is not calculated but pre-determined based on the Nyquist law. The data acquisition device should collect the data at the preset sampling rate. e.g. If the sampling rate is 200 s/s, so the data should be collected every 5 ms. The fs in the paper is an estimation value. Therefore, the peak frequency in Fig. 6 is not correct because it is related to fs.

The angles and the orientations of the accelerometer are not clearly defined in Fig.4. Where are the sides of the angles? Missing the description of the orientations of the x, y and z axes.

The standard deviation should be shown as the error bar in Fig. 7.

Author Response

Dear Honorary editors, reviewer

Thank you very much for so kindly reviewing our paper and making further very valuable constructive comments. We have considered all your comments carefully and have incorporated them into the revised paper, significantly enhancing it. The changes made are summarised in the attached table. We have also proofread the paper again.   

We are very grateful for the help and support provided and hope our revisions meet your expectations.

Best wishes

Professor Reza Saatchi

Round 3

Reviewer 2 Report

Figure 4 still did not show the relationship among the angles and the acceleration measured by the two accelerometers. What are the sides of angles ßthigh and φshank? Is the z direction of the accelerometers perpendicular to the femur and tibia? Is the y direction of the accelerometers parallel to the femur and tibia? The figure is to validate if the equations 2-5 are correct. It is unclear why the ROM is different in flexion-extension from walking.

The previous concern on the statement of sampling frequency was not addressed correctly. Sampling frequency should be determined before the data acquisition. It is the reference value and should be accurate. The experiment setup is not correct. It should sample the data at the predetermined rate rather than as fast as possible. It may well remove the statements of the sampling frequency as the result was independent to time. Figure 6 did not provide any information and the frequency is not accurate because of the accuracy of fs. It should be removed.

The statement "sample size is sufficient" at line 444 is too strong and should be deleted. 

Author Response

Dear Honorary editors, reviewer

Thank you very much for so kindly reviewing our paper and making further very valuable constructive comments. We have considered all your comments carefully and have incorporated them into the revised paper, significantly enhancing it. The changes made are summarised in the attached table and highlighted yellow on the paper. We have also proofread the paper again.   

We are very grateful for the help and support provided and hope our revisions meet your expectations.

Best wishes

Professor Reza Saatchi

Round 4

Reviewer 2 Report

The authors should keep figure 4, especially they have a reference figure in ref 42. They can redraw the figure to support the equation.

The authors misunderstood the comment regarding the sampling frequency. It is not about if the 244Hz is the right frequency. In fact, it is oversampling based on the signal measured. The issue is how the sampling frequency is determined. The sampling frequency is preset, not measured.  The description regarding the sampling frequency is against the data sampling protocol. It can be removed because the data is not related to time now. 

Author Response

Dear respected reviewer

Thank you very much for so kindly reviewing our paper and making further very valuable constructive comments. We have considered your comments carefully and have done our best to amend the paper accordingly. The changes made are summarised in the attached table and highlighted yellow on the paper.   

We are very grateful for the help and support provided and hope our revisions meet your expectations.

Best wishes

Professor Reza Saatchi
